# Understanding the Role of HLA Class I Molecules in the Immune Response to Influenza Infection and Rational Design of a Peptide-Based Vaccine

**DOI:** 10.3390/v14112578

**Published:** 2022-11-21

**Authors:** A. K. M. Muraduzzaman, Patricia T. Illing, Nicole A. Mifsud, Anthony W. Purcell

**Affiliations:** Infection and Immunity Program, Department of Biochemistry and Molecular Biology, Biomedicine Discovery Institute, Monash University, Clayton, VIC 3800, Australia

**Keywords:** influenza, HLA, vaccine

## Abstract

Influenza A virus is a respiratory pathogen that is responsible for regular epidemics and occasional pandemics that result in substantial damage to life and the economy. The yearly reformulation of trivalent or quadrivalent flu vaccines encompassing surface glycoproteins derived from the current circulating strains of the virus does not provide sufficient cross-protection against mismatched strains. Unlike the current vaccines that elicit a predominant humoral response, vaccines that induce CD8^+^ T cells have demonstrated a capacity to provide cross-protection against different influenza strains, including novel influenza viruses. Immunopeptidomics, the mass spectrometric identification of human-leukocyte-antigen (HLA)-bound peptides isolated from infected cells, has recently provided key insights into viral peptides that can serve as potential T cell epitopes. The critical elements required for a strong and long-living CD8^+^ T cell response are related to both HLA restriction and the immunogenicity of the viral peptide. This review examines the importance of HLA and the viral immunopeptidome for the design of a universal influenza T-cell-based vaccine.

## 1. Introduction

Whilst Hippocrates described influenza disease in 412 BC (reviewed in [1]), the first influenza A virus (IAV) pandemic, known as the Russian Flu, was not reported until 1889 [2]. This was followed by the isolation of the virus half a century later in 1933 [3]. Since then, IAV has affected a significant percentage of the world’s population through regular seasonal epidemics and occasional pandemics, causing millions of deaths [4]. The best way to limit influenza’s medical, social, and economic burdens is through annual vaccination, as the World Health Organization (WHO) recommended in their Global Action Plan [5]. Unfortunately, the current antibody-inducing trivalent or quadrivalent flu vaccines have several limitations, including their inadequate boosting of cellular immunity via cytotoxic T cell (CTL) responses [6,7,8,9,10]. Due to antigenic drift, which generates genetic changes or mutations in cell surface glycoproteins, hemagglutinin (HA), and neuraminidase (NA), the current influenza vaccines sometimes fail to produce neutralizing antibodies against seasonal or emerging influenza virus strains [11,12,13]. The United States Centers for Disease Control and Prevention (US-CDC) vaccine effectiveness study showed that the efficacy of the current strain-based vaccines can vary from as low as 10% in an unmatched season to 60% in the case of a very well-matched vaccine [14]. The currently available trivalent (composed of surface glycoproteins from two IAVs and one influenza B virus (IBV)) and quadrivalent (consisting of surface glycoproteins from two IAVs and two IBVs) vaccines require annual reformulation and administration to circumvent the effects of antigenic drift. In addition to humans, IAV can infect a wide range of animals, including pigs, horses, marine mammals, cats, dogs, and birds. Due to this wide host range, antigenically novel strains or IAV subtypes can emerge [15], triggering global pandemics, as observed in the last century (1918 Spanish flu, 1957 Asian flu, 1968 Hong Kong flu) and, more recently, the 2009 H1N1 pandemic (pdmH1N1) [2,16].

There is considerable evidence to suggest that, in both humans [17,18,19] and mice [20,21,22], CD8^+^ T cells are important in providing protection against influenza through viral clearance, thereby limiting the disease severity. Therefore, T-cell-inducing vaccines, particularly CD8^+^ CTLs, targeting the highly conserved internal proteins, may be an effective alternative for the provision of universal protection against influenza virus subtypes [23,24]. To stimulate the T cells, viral antigens are processed by the infected host cells and presented as short peptides by HLA molecules encoded within the human major histocompatibility complex (MHC) [25]. The human *MHC* is the most polymorphic gene system [26], and the encoded HLA allotypes display a diverse peptide array to the T cells for immune recognition (termed the immunopeptidome). Importantly, HLA polymorphisms have been shown to influence peptide binding specificity, T cell receptor (TCR) interactions with the peptide/HLA complex, and the nature of the ensuing anti-viral immune response. Studies shown that some HLA alleles, such as HLA-B*27, that present a conserved epitope from the p24 HIV capsid protein are protective, resulting in a significantly improved survival by maintaining high CD4 counts and a low-level viral load without antiretroviral therapy [27]. Conversely, some other HLA alleles, such as HLA-B*35, are associated with rapid progression and a high viral load [28]. HLA associations have also been well reported for chronic infectious diseases, including hepatitis B and hepatitis C [29,30,31]. However, associations with the influenza virus pathogenesis are still emerging. This review focuses on the identification of the contributions of classical HLA class I (HLA-I) allotypes to the immunoregulation of influenza’s pathogenesis and implications for the development of a universal T cell-inducing vaccine.

## 2. Influenza: An Unpredictable Virus with a High Pandemic Potential

Influenza viruses belong to the *Orthomyxoviridae* family and are classified into four subtypes: IAV, IBV, influenza C (ICV), and influenza D (IDV) [32,33,34]. All influenza viruses are enveloped and contain a negative-sense, single-stranded segmented RNA genome [32]. Although IAV and IBV virions contain eight viral RNA (vRNA) segments, ICV [35] and IDV [34] contain seven vRNA segments. A schematic representation of IAV/IBV is shown in Figure 1A. The influenza virion has three subviral components. The first is an envelope lipid bilayer incorporating three virally encoded transmembrane proteins, HA, NA, and the proton channel protein, matrix-2 (M2). Secondly, a matrix layer underneath the envelope, composed of M1, interacts with the cytoplasmic tails of HA, NA, and M2 on the outer side and vRNA and nucleoprotein (NP) on the inner side of this layer [36,37]. Thirdly, the viral core consists of helical ribonucleoprotein complex (RNP) segments containing the negative-stranded genomic vRNAs, NP, the nuclear export protein (NEP) and a 3-polymerase (3P) complex. The 3P complex is heterotrimeric, being composed of the polymerase basic protein 1 (PB1), polymerase basic protein 2 (PB2), and polymerase acidic protein (PA) [36,37,38,39,40].

Among the four subtypes of the influenza virus, IAV is the most important public health consideration in regard to seasonal epidemics and potential pandemic threats. In addition to humans, IAV can infect a large number of animal species, including ducks, chickens, pigs, whales, horses, and seals [41,42]. Different combinations of IAV HA (1–18) and NA (1–11) proteins can potentially give rise to 198 subtypes, although only 131 subtypes have been naturally isolated [41]. Different IAV subtypes, such as H5N1, H7N9, and H9N2, circulating in aquatic birds are called avian influenza viruses. These avian strains sometimes switch between hosts to form new lineages and may emerge as antigenically novel human IAVs that can promote pandemics [43]. Although human infections with avian influenza viruses are rare, this can occur following exposure to infected poultry, either via direct contact with chickens or by visiting a live poultry market [44,45,46,47]. In the last century, there have been four instances where influenza viruses with genes originating from avian or swine reservoirs emerged, either with or without reassortment with human influenza viruses, resulting in a pandemic wave [48]. As the IAV reservoirs cannot be eradicated from wild aquatic birds, another influenza pandemic is inevitable, and based on past experience, it is widely believed that the currently circulating avian influenza virus H5N1 or H7N9 may be the culprit subtype.

Influenza viruses are constantly evolving via two different mechanisms known as antigenic drift and antigenic shift (Figure 1B). Firstly, antigenic drift is the result of minor changes or mutations due to error-prone RNA replication, which occurs in both IAV and IBV as the virus replicates. Secondly, novel influenza viruses with pandemic potential can arise through the antigenic shift resulting from abrupt changes in, or the reassortment of, gene segments between avian or swine influenza viruses, which cross the species barrier and infect humans [49]. Due to these rapid antigenic drifts or shifts, vaccines against surface glycoproteins may be ineffective against seasonal or novel influenza viruses [9,50,51,52,53,54]. In contrast, influenza virus internal proteins (i.e., NP, PA, PB, NS and M) are highly conserved across the different subtypes [49]. Importantly, studies have shown that CD8^+^ T cells can recognize antigenic peptides derived from conserved internal proteins, including NP, M1, and PB1 [55,56,57,58]. A recent study has shown that memory CD8^+^ T cells specific to a conserved PB1 (A2-PB1_413–422_) peptide cross-reacted with homologous sequences from IAV, IBV, and ICV, offering a degree of cross-strain immunity [59]. It is recognized that the epitopes derived from conserved internal proteins are likely to elicit broader immunity against a spectrum of influenza strains so as to promote lifelong protection [60]. Thus, T cells against these conserved proteins have the potential to provide heterosubtypic protection against influenza disease and are of substantial interest for the design of next-generation universal vaccines.

## 3. Host Immunity to Influenza

When an individual is infected with influenza virus, both the innate and adaptive arms of the host immune system are engaged to combat and clear the pathogenic assault. The innate immune system is the first defense line that recognizes the viral components called pathogen-associated molecular patterns (PAMPs) by the host pathogen recognition receptors (PRRs). For example, in influenza virus infection, the retinoic-acid-inducible gene-I protein (RIG-I) recognizes the viral RNA in the cytosol and induces the production of pro-inflammatory cytokines and type I IFN. Toll-like receptor (TLR) 3 and TLR7 recognize double-stranded and single-stranded RNA (dsRNA and ssRNA) and many more PRRs which target different proteins of the influenza virus, also leading to the activation of innate immune signaling and the induction of cytokine and antiviral molecule expression [61,62,63]. However, the professional antigen-presenting cells, e.g., dendritic cells (DCs), link the innate and adaptive immune responses during the IAV infection [64]. After infection with IAV, the conventional DCs (cDCs) migrate from the lungs to the lymph nodes and present antigens derived from IAV with major histocompatibility complex (MHC) class I molecules for the purpose of recognition by virus-specific CD8^+^ CTLs [65,66]. In addition, viral proteins degraded in endosomes/lysosomes are associated with the MHC class II molecule and are presented on the cell membrane for their recognition by CD4^+^ T helper (Th) cells that leads to B cell proliferation and maturation into antibody-producing plasma cells [67,68]. The humoral immune response is mediated by B cells that produce neutralizing antibodies targeting viral surface proteins, such as HA and NA. These antibodies can also induce antibody-dependent, cell-mediated cytotoxicity or complement activation by directly binding to the surface of infected cells [52,69]. The cellular immune response is primarily mediated by CD8^+^ T cells, which play a crucial role in the surveillance, detection, and elimination of virus-infected cells (Figure 2) [70,71,72]. Indeed, T cells have been shown to play an essential role in the clearance of many respiratory viruses [73,74,75,76].

Animal studies have demonstrated the protective role of CD8^+^ T cells against IAV infection. When influenza virus-specific CD8^+^ T cells were injected into naïve mice, they showed an increased survival and decreased weight loss after being challenged with a lethal dose of virus, confirming the role of CTL in viral clearance and protection [77,78]. In contrast, T cell-deficient mice (athymic mice bred on a BALB/c background) or transgenic mice lacking MHC-I showed delayed viral clearance and increased mortality [71,79]. Most importantly, heterosubtypic immunity, wherein host immune responses to highly conserved peptides across all the influenza virus subtypes take place, confers protection against infection and disease caused by different subtypes. Although the evidence regarding humans is limited, T cell-mediated heterosubtypic immunity against influenza viruses has been confirmed in murine studies. When conserved NP-peptide-specific T cell clones (DbNP_366_ epitope) or H1N1-virus-specific T cell clones were transferred into naïve mice and then challenged with different IAV subtypes, these mice showed the early recruitment of T cells in the lung tissues, leading to early viral clearance and increased survival [70,80,81,82]. CTL heterosubtypic immunity has also been confirmed in guinea pigs, in which priming with the 2009 pandemic H1N1 virus conferred protection against the 2013 avian H7N9 IAV [83].

In humans, the control of influenza infection by the CD8^+^ T cells was first described by McMichael et al., who demonstrated that the IAV titers and clinical symptom severity were inversely correlated with the magnitude of the pre-existing anti-influenza CD8^+^ T cells [84]. Furthermore, studies found that amino acid changes in the major viral NP CTL epitopes (HLA-B*27:05-restricted epitope NP_383–391_ (SRYWAIRTR) and the HLA-B*08-restricted epitope NP_380–388_ (ELRSRYWAI)) [85] can rapidly accumulate in vivo, suggesting that CD8^+^ CTLs exert selection pressure on the virus [85,86,87,88,89].

The first evidence of heterosubtypic protection against a pandemic strain was described by Epstein in the Cleveland family study during the 1957 H2N2 pandemic [90]. The results showed that adults who had laboratory-confirmed H1N1 seasonal influenza virus infections between 1950 and 1957 were three times less likely to have symptomatic laboratory-confirmed pandemic H2N2 influenza virus compared to those who were not previously infected with the seasonal H1N1 influenza virus [90]. Recently, during the 2009 H1N1 influenza virus pandemic, individuals with pre-existing IAV-specific CD8^+^ T cells showed a decreased risk of fever, fewer influenza-like symptoms of illness, reduced illness severity scores, and the absence of viral shedding [17]. In addition, the flu watch cohort study also observed a positive correlation between pre-existing IAV-specific CD8^+^ T cells and less symptomatic IAV infections during three seasonal epidemics and the 2009 H1N1 pandemic, which was independent of baseline antibodies, confirming the heterosubtypic protection delivered by CD8^+^ T cells in influenza virus infection [19]. Many other studies have also described the potential cross-reactive response of the CD8^+^ T cells to different influenza virus strains, including the highly pathogenic H7N9 and H5N1 avian influenza viruses [18,56,82,91,92,93,94,95]. Indeed, H7N9-infected patients with early H7N9-specific, IFN-γ-producing CD8^+^ T cell populations recovered more rapidly from severe infection. In contrast, patients with delayed or absent CD8^+^ T cell responses had increased morbidity and mortality [18]. These studies, collectively, confirm the role of the CD8^+^ T cells in influenza recovery.

## 4. Immunodominance and MHC Genotype in the Antiviral Response

Immunodominance is a central feature of immune responsiveness to viruses. It refers to the phenomenon whereby the antiviral CTL population tends to be dominated by T cells restricted to a small number of antigenic MHC-peptide complexes (i.e., presenting immunodominant epitopes) among the available MHC peptides [96]. Indeed, immunodominant epitopes have been shown to play a critical role in eliminating virus-infected cells and contributing to the development of a memory T cell pool so as to combat subsequent encounters [96,97,98,99]. The mechanism of immunodominance is complex, involving features associated with (i) epitope abundance, (ii) the proteasomal processing of antigens, with immunoproteasomes playing an important role in the establishment of the immunodominance hierarchy of CD8^+^ T cells [100], (iii) IFN-γ production [101], (iv) TCR repertoire and affinity [102], and (v) the structural features of the antigen (disulfide bonds, protease-processing sites), all of which contribute to the overall CTL response [103,104,105,106,107,108,109]. A number of studies demonstrated that the co-expression of different MHC (or HLA) allotypes influences both the magnitude and specificity of the CTL response to influenza infection. Peripheral blood mononuclear cells (PBMCs) isolated from an individual co-expressing HLA-A2 and HLA-B37 were stimulated with either infectious virus, influenza A (A/PR8 (H1N1)), or synthetic peptides, HLA-A2-restricted M_57–68_ and HLA-B37-restricted NP_335–349_, and only an HLA-B37-restricted CTL response was observed [110]. Although the immunodominance of the HLA-A2–restricted response to the conserved IAV M57-68 peptide has been well-established [111,112], the immunodominance of NP_335–349_ over the HLA-A2-restricted M_57–68_ was observed in individuals co-expressing HLA-A2 and B37. In two separate studies, Boon et al. observed that the magnitude of influenza-A-virus-specific CTL responses depended on the HLA-A and -B alleles expressed [112,113]. Here, the frequency of HLA-B8-restricted NP_380–388_-specific CTLs were 3-fold lower in HLA-B27^+^ carriers compared to HLA-B27^-^ individuals (*n* = 5). Additionally, HLA-A1-restricted NP_44–52_-specific CTLs were 3-fold higher in individuals (*n* = 4) that co-expressed HLA-A1, HLA-A2, HLA-B8, and HLA-B35, which indicated that the overall CTL response was influenced by an individual’s complete HLA genotype [112]. In their second study, it was demonstrated that both HLA-B*08:01 and HLA-A*01:01 influenza-A-virus-specific CTL responses were much higher in HLA-B*27:05^+^ and HLA-B*35:01^+^ individuals [113]. Indeed, in the presence of HLA-B*27:05, the magnitude of the HLA-B*08:01-restricted response was even lower [113]. The effects of different HLA genotypes on immunodominant peptide presentation was examined by Wu et al. [57]. After comparing the immunogenicity towards the influenza M_58–66_ epitope in healthy HLA-A*02:01^+^ donors, they reported that the response was only immunodominant in three out of eight individuals. The remaining five HLA-A2^+^ carriers demonstrated a prominent CD8^+^ T cell response to the NP_380–388_ epitope that was restricted to HLA-B*08:01^+^ [57]. Indeed, HLA-B allotypes have been shown to significantly (*p* = 0.012) generate a more robust polyfunctional (via the production of IFN-γ and IL-2) CD8^+^ T cell response to different viral antigens, encompassing HIV-1, CMV, EBV, and the influenza virus, compared to HLA-A allotypes [114].

Additionally, Quiñones-Parra et al. demonstrated that HLA-A*03:01 NP_265–273_, HLA-B*08:01 NP_225–233_, HLA-B*18:01 NP_219–226_, HLA-B*27:05 NP_383–391_, and HLA-B*57:01 NP_199–207_ epitopes were capable of eliciting robust CTL responses to any IAV strain in individuals carrying these HLA allotypes. Contrary to this, limited CTL responses were observed in the ethnicities in which HLA-A*01:01, HLA-A*24:02, HLA-A*68:01, and HLA-B*15:01 are prominent [115]. Moreover, HLA-B*37:01 showed promising results in presenting different NP peptide variants capable of producing robust CTL reactivity to protect against H1N1, pH1N1, H3N2, H5N1, H7N9, and possibly other novel IAV strains [116]. Furthermore, HLA-A*02:01 PB1_413–421_ can exhibit universal cross-reactivity across IAV, IBV, and ICV, and HLA-A*01:01 PB1_591–599_ and HLA-B*37:01 NP_338–345_ peptides could exhibit cross-reactive CD8^+^ T cell responses in IAV and IBV infections [59].

A recent study reported that serologically defined HLA-A2 homozygous lymphocytes, in contrast to heterozygous lymphocytes, did not synthesize detectable influenza virus RNA or protein after exposure to the virus. Meanwhile, the HLA-A2 homozygous lymphocytes could internalize the infectious viruses, and they were not susceptible to lysis by autologous virus-specific CTLs. Similar findings were also observed in the case of HLA-A1, -A11, and -B44 homozygous lymphocytes, suggesting that individuals carrying homozygous HLA class I allotypes may possess lymphocytes that are not susceptible to influenza virus infection and CTL lysis, which is a unique HLA-associated lymphocyte defense mechanism against influenza virus [117]. Therefore, the combination of HLA allotypes expressed by an individual may impact peptide presentation, the modulation of surface expression, and the competition for overlapping peptides (this can lead to a decrease in the ability to generate a CTL response). Moreover, HLA carriage can also shape the activation and proliferation of T cells during infection [112,118,119,120,121,122,123,124].

## 5. HLA Association with Influenza Disease Susceptibility: Population-Based Studies

Population studies examining the HLA-associated influenza pathogenesis provide critical information regarding disease protection or susceptibility. Global indigenous population studies have reported higher incidences of morbidity and mortality during both seasonal and pandemic influenza virus infections than in non-indigenous populations [125,126,127,128,129,130,131,132,133,134,135,136]. The lesser degree of HLA diversity exhibited by indigenous communities is likely to have arisen from an evolutionary bottleneck that established a small ancestral pool with little HLA diversity. This evolutionary effect is amplified by limited outbreeding to other populations, which may explain the low prevalence of protective HLA (e.g., only ~15% HLA-A2 among the Indigenous peoples of Australia) against influenza virus infection [135]. To analyze the preferential antigen binding and capability of the T cell responses of different HLA alleles, Hertz et al. introduced a computational prediction tool called the “targeting efficiency”. This tool analyzes the Spearman correlation coefficient of a given HLA molecule binding to the conserved regions of a protein [137]. When an HLA preferentially binds the conserved regions, it provides a positive score, and a negative score indicates the preferential binding of variable regions. To determine whether the targeting efficiency is correlated with the CD8^+^ T cell response in influenza virus infection, thirteen pH1N1-positive individuals were recruited in the 2009–2010 influenza season and ranked based on their average HLA-A and HLA-B allele targeting efficiency scores over the entire pH1N1 proteome. The enzyme-linked immunosorbent spot (ELISpot) was used to detect IFN-γ secretion patterns from PBMCs and showed that the HLA targeting efficiency scores were correlated with IFN-γ spot-forming units (SFUs) in regard to pH1N1 (*r* = 0.54, *p* = 0.054) [134]. To assess the broader implications of the targeting efficiency on the population level, 95 of the most prevalent HLA class I alleles against a wide variety of influenza virus subtypes, including 2009 pH1N1, previously circulating H1N1 and H3N2 subtypes, and H5N1 and 1918 H1N1 subtypes, were computed. The study identified three HLA alleles (HLA-A*24, HLA-A*68, HLA-A*39) which had a low targeting efficiency score and, hence, were positively correlated with pH1N1 mortality. The high prevalence of HLA-A*24 in the Indigenous peoples of Australia (where HLA-A*02:01 (17%), HLA-A*11:01 (16%), HLA-A*24:02 (22%), and HLA-A*34:01 (24%) account for 79% of the HLA-A allotypes) may underpin the vulnerability of this population in an influenza virus pandemic or even cases of seasonal influenza [135]. Another study aiming to examine the susceptibility to avian influenza virus (H7N9) among different ethnicities identified six conserved epitopes from the matrix and NP proteins of H7N9 influenza virus, which are restricted to HLA-A*03:01(NP_265_^+^), HLA-A*02:01(M1_58_^+^), HLA-B*27:05 (NP_383_^+^), HLA-B*57:01(NP_199_^+^), HLA-B*18:01(NP_219_^+^), and HLA-B*08:01(NP_225_^+^), that displayed robust CD8^+^ T cell responses. Unfortunately, the identified HLA alleles that produced robust CD8^+^ T cell responses were found at a very low frequency in the Alaskan and Australian Indigenous populations (16%) compared to higher frequency in other ethnicities (African, 37%; European descent, 57%; Oriental, 37%; Amerindian, 36%). Thus, the influenza disease severity could therefore show an ethnic bias in the face of an H7N9 pandemic [115].

Different HLAs may also be more efficient in presenting specific influenza virus proteins. Most studies have shown that the NP, M, and PA are the sources of most T cell epitopes in both humans and mice [58,111,138,139,140]. This result may have some bias, as these influenza proteins were mostly studied in the context of the most prevalent HLA alleles, such as HLA-A2. Even then, when Hensen L et al. identified the immunopeptides of IAV by LC-MS/MS restricted to HLA-A*24, they discovered that the majority (62%) of the peptides are from PB1 and PB2, with none derived from the NA or M proteins [141]. These findings highlight the importance of the including PB1- and PB2-derived peptides in a universal influenza vaccine design so as to cover vulnerable global Indigenous populations [141].

In a case–control study in the Mexican-Mestizo population, a lower frequency of HLA-A*24:02:01 in patients compared to asymptomatic contacts suggested a possible protective effect [142], which conflicts with the low targeting efficiency of HLA-A*24 described by Hertz et al. [134]. The same study also identified the HLA-A*68:01:01 as a susceptible allele, as it was found only in the influenza-positive patient group [142], which, again, corelates with the low HLA targeting efficiency score of HLA-A*68:01 but not the closely related allotype, HLA-A*68:02 [134]. The examination of the HLA-B allotypes in the same Mexican population revealed HLA-B*39:06:02 and HLA-B*51:01:05 as susceptible allotypes, which were observed at a high frequency in individuals who experienced the 2009 influenza A H1N1 [142]. Additionally, a Northeast Indian study reported that, during the 2009 pandemic, the HLA-A*11 and HLA-B*35 allotypes conferred susceptibility to IAV (H1N1) [143]. In contrast to these influenza-susceptible alleles, different studies have shown that HLA-A*01:01, HLA-A*02:01, HLA-A*03:01, HLA-B*08:01, HLA-B*18:01, HLA-B*27:05, HLA-B*37:01, and HLA-B*57:01 present highly conserved influenza viral peptides and were associated with universal protective CD8^+^ T cell-mediated immunity [59,115,144]. Although not statistically significant, a higher prevalence of HLA-C*03:02 (3.98% in the influenza positive group compared to 1% in the influenza contact group) was observed in the Mexican IAV H1N1/09 patient group. In contrast, HLA-C*03:03 (1.44% patients vs. 5.44% contact), HLA-C*03:04 (4.34% patients vs. 10.66% contact), and HLA-C*07:01 (5.07% patients vs. 9.33% contact) were reported at lower frequencies in the patient group [142]. There is minimal information regarding the influenza viral epitopes presented by HLA-C, with only 30 entries in the Immune Epitope Database (IEDB) [145]. Interestingly, compared to the known GILGFVFTL–HLA-A*02:01 structure (PDB 2VLL), the structure of GILGFVFTL–HLA-C*08:01 (PDB 4NT6) demonstrated near-identical conformations and elicited CTL responses in both HLA-A*02- and HLA-C*08-positive individuals [141]. This finding provided evidence for the degenerate antigenicity between HLA products derived from separate loci and could provide new insights for the future development of universal IAV vaccines.

## 6. Towards a Universal T Cell Epitope Vaccine

The most recent pandemic of severe Acute Respiratory Syndrome Coronavirus 2 (SARS-CoV-2) and the rapid emergence of novel variants have reminded us how crucial it is to possess an effective vaccine when the treatment options are limited. The ideal and most effective vaccine is likely to employ both cellular and humoral immune responses. Unfortunately, the appearance of the variants of viruses such as SARS-CoV2, IAV, or some persisting viral infections, such as HIV and hepatitis C, that evade vaccines or pre-existing antiviral immunity has often raised the issue of the use of antibody-inducing vaccines alone. The T cells play a critical role in the duration and cross-reactivity of vaccines. Moreover, pre-existing T cell immunity is associated with the decreased severity of infectious diseases.

The most significant advantage of a T cell-inducing influenza vaccine is based on its cross-protective capability against different strains, engendered by conserved epitopes. Substantial progress has been made in the identification of conserved peptides that generate cross-protective CTL responses across all three influenza virus subtypes [59] or individual peptides that can generate a robust immune response in vulnerable indigenous populations [141]. Still, there is a considerable knowledge gap in regard to the formulation of a universal vaccine due to the complexity of HLA diversity and restricted T cell responses.

In the last ten years, more than 20 T cell-based vaccines have been in various stages of clinical development, with the majority of studies focusing on NP and M1 epitopes using different vaccine delivery platforms (Table 1). Notably, most of the vaccines have shown a promising safety and efficacy via the induction of CD8^+^ T cell responses. However, no candidate vaccine has proven to be effective in HLA-diversified populations. Although some data are available regarding specific HLA (especially HLA-A2), there is limited information on the magnitude, quality, and breadth of influenza virus-specific T cell responses in African, Latin American, and Asian populations. Therefore, we must map the conserved antigenic epitopes across the different HLAs prevalent in diverse populations to design a genuinely universal influenza vaccine. It is likely that there will need to develop different formulations so as to address the HLA diversity across global populations, including the highly vulnerable indigenous people.

## Figures and Tables

**Figure 1 viruses-14-02578-f001:**
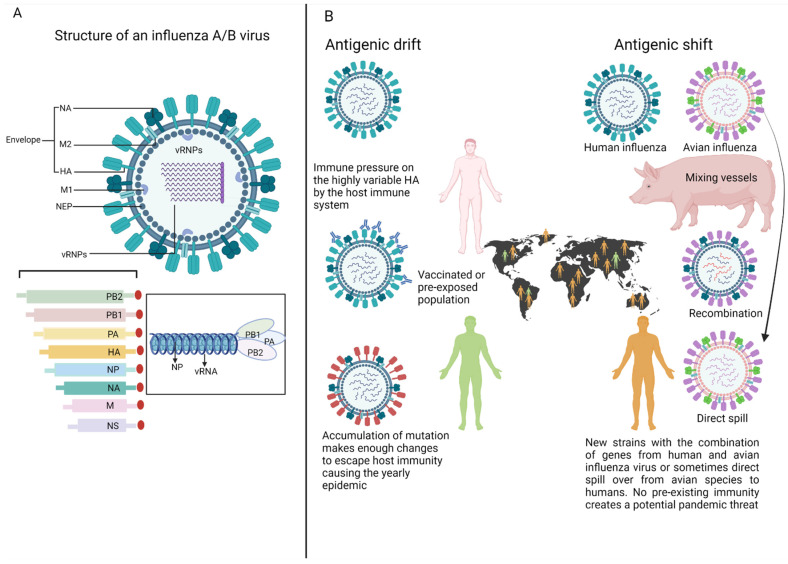
Influenza virus structure and antigenic drift and shift in influenza infections. (**A**) Typical influenza virus spheroidal form showing all the viral components. HA, NA, and M2 proteins form the envelope, with M1 forming a critical bridge between the viral envelope and the vRNP core, which consists of helical RNP segments containing negative-stranded genomic vRNAs and NP, along with the 3-P complex. (**B**) Over time, small changes associated with antigenic drift can accumulate and result in viruses that are antigenically different from their ancestors. The current vaccines that induce strain-specific antibodies do not afford complete protection, and as a result, we observe yearly epidemic outbreaks of seasonal influenza. Antigenic shift can result in a new influenza A subtype which is either generated via an intermediate host where there is, for example, gene exchange between human and avian influenza viruses, forming a new strain capable of infecting humans, or, sometimes, influenza strains from other species can directly infect humans by crossing the species barrier (e.g., in the case of H5N1). When antigenic shift occurs, most people have little or no immunity against the new virus, thereby enabling pandemics. Image created with BioRender.com. (accessed on 14 September 2022).

**Figure 2 viruses-14-02578-f002:**
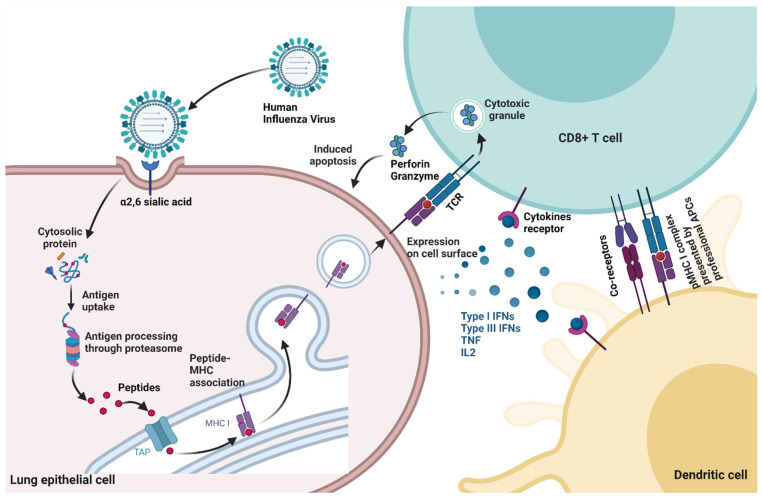
T cell recognition of influenza virus-infected cells. Upon infection, respiratory epithelial cells and professional antigen-presenting cells (APCs), such as dendritic cells (DCs), process the viral proteins and present them as peptides in complex with MHC-I on the cell surface. Specific T cell receptors (TCR) expressed on the CD8^+^ T cells then recognize these peptide/MHC-I complexes. Upon engagement, the CD8^+^ T cells become activated and secrete cytotoxic granules (i.e., granzymes, perforin) targeting the infected cell to induce its death and produce proinflammatory cytokines (i.e., IFNs, TNF, and IL2). Image created with BioRender.com. (accessed on 14 September 2022).

**Table 1 viruses-14-02578-t001:** Universal vaccine candidates in clinical trials (December 2021).

Vaccine Name	Target	Platform	Targeted Age Group	Adjuvant	Phase and Major Findings	Reference
MVA-NP+M1	NP, M1	Viral vector	50 and over	None	Phase I: Safe and induces NP- and M1-specific CD8+IFNγ+ T cells	[146,147,148]
					Phase IIa: Increased T-cell-killing capacity Reduced duration of virus shedding	[149,150]
					Phase IIb: Completed. Result to be published	[151,152]
M-001	HA, NP, M1	Recombinant protein	65 and over	Montanide ISA-51	Phase I/II: Well-tolerated and safe Enhanced seroconversion Significant elevation in influenza-specific CMI	[153]
					Phase II: Induced broad protective T cells effective against different strains	[154]
					Phase III: Results submitted on 16th June 2021	[155,156]
FLU-v	NP, M1, M2	Peptide-based	18–40	Montanide ISA-51	Phase I: Safe and well-tolerated IFN-γ responses > 2-fold	[157]
					Phase II: Well-tolerated IFN-γ 38.2-fold TNF-α 7.0-fold IL-2 1.7-fold	[158]
VGX-3400X	HA, NA, NP	DNA	Unknown	None	Phase I: Not available	[159]
FP-01.1	NP, M1, P1, P2	Peptide-based	22–55	None	Phase I Acceptable safety and tolerability Generated robust anti-viral T cell responses	[160,161,162,163,164]
ChAdOx1-NP+M1	NP. M1	Viral vector	18–50	None	Phase I: Highly immunogenic and well-tolerated	[165]
VXA-A1.1	HA	Viral Vector	18–49	TLR Agonist (dsRNA)	Phase I: Well-tolerated 4-fold HAI	[166]
					Phase II: 48% of participants were protected	[167]
Avian influenza virus VLP	HA NA	Viral vector		VLP	Phase I: Not available	[168]
MER4101	Whole virus	Attenuated virus			Phase I: Ongoing	
VAL-506440	HA	Lipid nanoparticle	18–64		Phase I: H10N8–78.3% (HAI) and 87.0% (MN) H7N9–96.3% (HAI) and 100% (MN)	[169]
M2SR	M2	Attenuated virus			Phase I: Ongoing	[170]
Immunose Flu	HA Whole virus	Split virion			Phase I/II: Completed Results not yet published	[171]
NanoFlu					Phase I/II: Completed Results not yet published	[172]
					Phase III: Ongoing	[173]
cH8/1N1, H5/1N1	HA stalk	Attenuated virus	18–39	ASO3A	Phase I: High-level cross-reactive serum IgG antibodies.	[174]
VRCFLUDNA081-00-VP	HA	Ferritin Nanoparticle			Phase I: Ongoing	[175]
OVX836	NP	Recombinant protein	18–65		Phase I: 10-fold HAI titre	[176]
					Phase II: Ongoing	[177]

## Data Availability

Not applicable.

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
