# Peer review of "Understanding the Role of HLA Class I Molecules in the Immune Response to Influenza Infection and Rational Design of a Peptide-Based Vaccine"

_viruses, 2022, doi:10.3390/v14112578_

Round 1
Reviewer 1 Report
Comments: This paper introduces the role of HLA class I molecules in the immune response to influenza infection and how to design an universal influenza T cell-based vaccine. The review is very comprehensive and interesting for guiding to develop universal influenza T cell-based vaccines. However, some minor concerns in the manuscript should be improved as follows.
1. In Abstract: “contagious” is not suitable, due to some subtype such as H9N2 is aerosol transmission.
2. In page 3, “These avain strains sometimes may”, “avain” is changed into “avian”. “Although human infections with avian influenza are rare”, “avian influenza” is changed into “avian influenza virus”, because avian influenza is a disease not virus, and the same problem also appear other place, such as in page 4, “When an individual is infected with influenza both”.
3. In figure 2, your described influenza virus entry into cells via α2,6 sialic acid, however, influenza virus also included avian influenza virus, which is via α2,3 sialic acid. And other question, why MHC I complex form in the lung epithelial cells but not dendritic cells.
4. In the part of universal vaccine candidates in clinical trials, the authors also describe the universal vaccines via humoral immunity, could you compare the advantages of T cellular immunity over humoral immunity though clinical data?
5. It is better to add a paragraph related the outlook for T-cell vaccine design.
6. Please add some references published in 2022.
Author Response
Comment-1
In Abstract: “contagious” is not suitable, due to some subtype such as H9N2 is aerosol transmission.
Response to Comment-1: We have removed the word contagious.
Comment-2
In page 3, “These avain strains sometimes may”, “avain” is changed into “avian”. “Although human infections with avian influenza are rare”, “avian influenza” is changed into “avian influenza virus”, because avian influenza is a disease not virus, and the same problem also appear other place, such as in page 4,“When an individual is infected with influenza both”.
Response to comment-2:
Changes have been made where necessary throughout the manuscript.
Comment-3
In figure 2, your described influenza virus entry into cells viaα2,6 sialic acid, however, influenza virus also included avian influenza virus, which is via α2,3 sialic acid. And other question, why MHC I complex form in the lung epithelial cells but not dendritic cells.
Response to comment-3
We have changed influenza virus to human influenza virus. Also, there were already DC cells on the right side of the figure presenting MHC I peptide complex to CD8+ T cells. We have labelled the figure as “pMHC complex presenting by professional APCs”.
Comment-4
In the part of universal vaccine candidates in clinical trials, the authors also describe the universal vaccines via humoral immunity, could you compare the advantages of T cellular immunity over humoral immunity though clinical data?
Response to comment-4
Thanks for picking up a very good point again. After careful review, we found that those HA-based vaccines are strain specific. So, it is irrelevant in the list of universal/T cell-based vaccines list. We have removed those from the list.
Comment-5
It is better to add a paragraph related the outlook for T-cell vaccine design.
Response to comment-5
We have added a small paragraph on the importance of T-cell-inducing vaccines. Discussing the designs of T cell-based vaccines is out of scope for this review.
Comment-6
Please add some references published in 2022.
Response to comment-6
We searched for related papers in 2022. We have added two recent 2022 publications for the universal vaccine design section.
Reviewer 2 Report
In this review, Muraduzzaman et al. focus on the current understanding of the role of HLA class I molecules in the immune response to infection of Influenza virus, which would be essential for the development of a universal T cell inducing vaccine. The manuscript is well structured and provides useful references to readers. I only suggest some minor revisions:
Introduction:
- A universal T cell inducing vaccines -> a universal T cell inducing vaccine
In Section “Immunodominance and MHC genotype in the antiviral response”
- use the same form for B27-negative and B27- (or positive and +)
In Section “HLA association with influenza disease susceptibility: population-based studies”
- The authors cited several studies arguing potential association between higher incidence of morbidity and mortality of influenza and lower HLA diversity and frequencies of specific HLA alleles observed in some indigenous populations. However, several recent studies also show that the HLA system might be more resistant to bottleneck effect compared to other genes and the potential presenting repertoire at population level is more or less maintained when several class I genes are considered together (see Buhler et al 2016 doi: 10.1007/s00251-016-0918-x), and patterns divergent allele advantage and functional complementarity between HLA genes have been suggested (Pierini & Lenz 2018 doi.org/10.1093/molbev/msy116, Di et al 2021 doi.org/10.1093/molbev/msaa325). A loss of HLA class I diversity in some population or the change of frequencies of some specific HLA alleles would not explain satisfactorily the more severe outcome of influenza in some populations.
- coefficient of the of a given -> coefficient of a given
- When a HLA preferentially -> When an HLA preferentially
- Keep using a same form for HLA alleles. For example: 02:01 -> A*02:01, 11:01-> A*11:01, HLA-A*02:01->A*02:01, -A*03:01 -> A*03:01, etc.
- Study 143: p=0.054 not significant
- Avoid the obsolete term “Caucasoid”. “European descents” would be better
References:
Please check the following references with wrong author names (part of some organisation or company names was apparently recognised as family names by software such as Endnote):
- Ref 68: GeurtsvanKessel
- Ref 173: Inovio Pharmaceuticals
- Ref 182: Services, N.D.o.H.a.H.
- Ref 183: Limited., N.I.L.M.P.I.N.L.
- Ref 186: AB., E.V.
- Ref 190: (CC), N.I.o.A.a.I.D.N.N.I.o.H.C.C.
and some others
Author Response
Comment-1
- A universal T cell inducing vaccines -> a universal T cell inducing vaccine
Response
Changed as suggested.
Comment-2
In Section “Immunodominance and MHC genotype in the antiviral response”
- use the same form for B27-negative and B27- (or positive and +)
Response
Changed as B27+ or B27- in the related section.
Comment-3
In Section “HLA association with influenza disease susceptibility: population-based studies”
- The authors cited several studies arguing potential association between higher incidence of morbidity and mortality of influenza and lower HLA diversity and frequencies of specific HLA alleles observed in some indigenous populations. However, several recent studies also show that the HLA system might be more resistant to bottle neck effect compared to other genes, and the potential presenting repertoire at the population level is more or less maintained when several class I genes are considered together (see Buhler etal 2016 doi: 10.1007/s00251-016-0918-x), and patterns divergent allele advantage and functional complementarity between HLA genes have been suggested (Pierini & Lenz 2018doi.org/10.1093/molbev/msy116, Di et al 2021doi.org/10.1093/molbev/msaa325). A loss of HLA class I diversity in some population or the change of frequencies of some specific-HLA alleles would not explain satisfactorily the more severe outcome of influenza in some populations.
Response
We agree with the reviewer that there may well be more complex explanations for more severe influenza outcomes in some populations than the simple model we discuss. Indeed, this could form a whole new topic looking at susceptibility to influenza infection and HLA. Given space constraints and the focus on identifying conserved T cell determinants we feel additional expansion of this small area of discussion in our paper is out of scope.
Comment-4
- coefficient of the of a given -> coefficient of a given
Response
Changed as suggested
Comment-5
- When a HLA preferentially -> When an HLA preferentially
Response-
Changed.
Comment-6
- Keep using a same form for HLA alleles. For example: 02:01 ->A*02:01, 11:01-> A*11:01, HLA-A*02:01->A*02:01, -A*03:01 ->A*03:01, etc.
Response
Changed HLA-A* or HLA-B* as a common format when describing HLA genotypes.
Comment-7
- Study 143: p=0.054 not significant
Response
We used the word correlated but did not use the word significantly correlated. Also, we have provided the P-value so that the reader would understand.
Comment-8
- Avoid the obsolete term “Caucasoid”. “European descents” would be better
Response
Changed as suggested.
Comment-9
References
Response
Carefully checked all the references and made changes as required.